# A New In Vivo Zebrafish Bioassay Evaluating Liver Steatosis Identifies DDE as a Steatogenic Endocrine Disruptor, Partly through SCD1 Regulation

**DOI:** 10.3390/ijms24043942

**Published:** 2023-02-15

**Authors:** Hélène Le Mentec, Emmanuelle Monniez, Antoine Legrand, Céline Monvoisin, Dominique Lagadic-Gossmann, Normand Podechard

**Affiliations:** 1INSERM, EHESP, IRSET (Institut de Recherche en Santé Environnement et Travail)-UMR_S 1085, University of Rennes, 35000 Rennes, France; 2UMR 1236-MOBIDIC, INSERM, Université Rennes, Etablissement Français du Sang Bretagne, 35043 Rennes, France

**Keywords:** zebrafish larvae, bioassay, endocrine disruptor compounds, NAFLD, liver steatosis, DDE

## Abstract

Non-alcoholic fatty liver disease (NAFLD), which starts with liver steatosis, is a growing worldwide epidemic responsible for chronic liver diseases. Among its risk factors, exposure to environmental contaminants, such as endocrine disrupting compounds (EDC), has been recently emphasized. Given this important public health concern, regulation agencies need novel simple and fast biological tests to evaluate chemical risks. In this context, we developed a new in vivo bioassay called StAZ (Steatogenic Assay on Zebrafish) using an alternative model to animal experimentation, the zebrafish larva, to screen EDCs for their steatogenic properties. Taking advantage of the transparency of zebrafish larvae, we established a method based on fluorescent staining with Nile red to estimate liver lipid content. Following testing of known steatogenic molecules, 10 EDCs suspected to induce metabolic disorders were screened and DDE, the main metabolite of the insecticide DDT, was identified as a potent inducer of steatosis. To confirm this and optimize the assay, we used it in a transgenic zebrafish line expressing a blue fluorescent liver protein reporter. To obtain insight into DDE’s effect, the expression of several genes related to steatosis was analyzed; an up-regulation of scd1 expression, probably relying on PXR activation, was found, partly responsible for both membrane remodeling and steatosis.

## 1. Introduction

Non-alcoholic fatty liver disease (NAFLD) is a growing worldwide epidemic responsible for an increasing number of chronic liver diseases and consecutive mortality [1]. The global prevalence of NAFLD is around 25% of the general population worldwide and could rise to 60–90% in some subpopulations, particularly those exhibiting metabolic diseases such as type 2 diabetes mellitus (T2DM) and obesity [1]. NAFLD covers a large panel of liver diseases, starting with liver steatosis characterized by an excessive accumulation within hepatocytes of fatty acids (mainly as triglycerides) inside lipid droplets. Although steatosis is considered a benign liver disease, this stage sensitizes the liver to subsequent harmful aggressions; thus, in 20% of cases, steatosis progresses to an inflammatory pathological state called non-alcoholic steatohepatitis (NASH), thereby predisposing toward more severe and irreversible complications such as cirrhosis and/or hepatocellular carcinoma [2,3,4]. However, the factors inducing the pathological development and progression and the underlying molecular mechanisms are not yet fully understood and remain to be further elucidated in order to better prevent the deleterious effects of the advanced stages of NAFLD.

Genetic predisposition, dietary habits, and metabolic disorders such as obesity and T2DM are the main risk factors of NAFLD. More recently, it has been acknowledged that exposure to environmental factors may also promote NAFLD [5], thus leading to the concepts of TAFLD and TASH (Toxicant-Associated Fatty Liver Diseases and Toxicant-Associated Steatohepatitis) [6,7,8]. Among environmental factors, endocrine disrupting compounds (EDCs), also called metabolism disrupting chemicals (MDCs), have been described for their ability to disrupt metabolic functions involved in the development of hepatic steatosis and the progression to steatohepatitis [9,10,11,12,13].

EDCs encompass a heterogeneous group of chemicals (plastics, plasticizers, pesticides, industrial solvents, heavy metals…) that have been mass produced throughout the past four decades, driven by their widespread use. Direct measurements of EDC levels in human blood and urine showed ubiquitous exposure to EDCs, through diet, lifestyle, and air contamination [14]. EDCs can interfere with any aspect of the endocrine system, including hormone production, release, transport, metabolism, receptor binding and/or activation. Notably, several EDCs can interact with nuclear receptors such as PPAR, CAR, PXR, LXR, or AhR, thus altering lipid metabolism and inducing the initiation or progression of TAFLD [10,15]. Among them, perfluorinated compounds, PCBs, phthalates (dibutyl phthalate [DBP], di(2)ethylhexyl)phthalate [DEHP]), dioxins (e.g., TCDD), BPA, and DDE (dichlorodiphenyldichloroethylene), the main metabolite of the insecticide DDT, have been associated with the development of steatosis in liver [9,16,17,18]. Although regulatory measures have been taken by the EU to restrict the use of certain EDCs (such as BPA and DDE), some of them have been replaced by substitutes whose safety has not been demonstrated (e.g., bisphenol S (BPS) and bisphenol F (BPF) as substitutes to BPA). In addition, due to the large number of new chemicals and to the lack of readily accessible and validated assays, it is still difficult to fully establish their potential health impacts, notably regarding the liver [19]. Therefore, there is a need to develop toxicological bioassays/screening methods to identify potential EDCs disturbing liver lipid metabolism, thus affording support to risk assessment of EDCs in relation to TAFLD.

In this context, the in vivo zebrafish larva model offers numerous advantages: complexity and variety of cell and organ interactions compared to in vitro models [20,21,22,23]; small size, transparency, rapid development (the liver is functional 3 days post-fertilization, dpf [22]); liver similarities to human with regard to functions and sensitivity towards xenobiotics (metabolism, toxicity, cellular and molecular responses) [21,24,25,26,27,28,29]; endocrine system similarities to human (nuclear/metabolic receptors mediating EDC effects and endocrine disruption) [30]. Concerning NAFLD and TAFLD, it has been shown that zebrafish can, in response to environmental triggers, develop hepatic steatosis [26,31,32,33,34] or transition from steatosis to steatohepatitis [35,36]. Moreover, in recent years, the zebrafish model at the embryonic and larval stages has become one of the most popular vertebrate models used for high-throughput drug screening and for studying lipid metabolism disorders and related diseases including NAFLD [34,37,38,39], notably in relation with EDCs (e.g., tributyltin [40,41]; perfluorooctane sulfonate [33]). Furthermore, according to the current European Union Animal Welfare Regulation (EU 2010), zebrafish younger than 5 days post-fertilization is not regulated by animal welfare legislation [42], and is thus considered an alternative model to animal experimentation. Based on all those properties, the zebrafish larva model is therefore an easily accessible, alternative experimental model for the in vivo screening of EDCs for their potential involvement in the development of liver steatosis and its transition to steatohepatitis.

In the present study, we developed an in vivo assay based on the quantification of lipid accumulation by fluorescent image analysis (Steatosis Assay on Zebrafish, StAZ) that highlights the induction of steatosis by EDCs. Using this bioassay, we investigated the effects of a panel of 10 selected endocrine disruptors on the induction of NAFLD. This panel of ten EDCs (Table 1) was established based on their relevance to human exposure, potential toxicity, and metabolism-disrupting properties (including obesity, diabetes, and NAFLD). Most of them are of regulatory interest in Europe (e.g., BPA, DEHP), and some are well-known EDCs (e.g., heavy metals, DDE], while several substitutes were also selected due to the lack of knowledge on their health effects (e.g., BPS and BPF). The main result of our EDC screening demonstrated that DDE is able to induce steatosis. This result was confirmed using an optimized StAZ test based on the use of a transgenic zebrafish model expressing a blue fluorescent protein in liver.

In an attempt to decipher the mechanisms involved in the initiation of steatosis caused by DDE, the mRNA expression of lipid metabolism-related genes was assessed. Our results suggest an increase of de novo lipogenesis as the main pathway of DDE-induced steatosis, notably linked to an increase of stearyl-CoA-desaturase 1 (scd1) expression that was found to be partly responsible for both membrane remodeling and liver steatosis.

## 2. Results

### 2.1. Development and Validation of the Steatogenic Assay on Zebrafish and Screening of EDCs for Their Capacity to Induce Steatosis

To address the need by regulatory instances of fast and reproducible tests to evaluate chemicals, we developed a test using zebrafish larvae with the aim to screen the potential of EDCs (or other chemicals) to induce steatosis. This bioassay, which we called Steatogenic Assay on Zebrafish (StAZ), is based on the fluorescence imaging of liver lipid droplets after Nile red staining following exposure of zebrafish larvae between 3 to 5 dpf (see Figure 1 for the flowchart of the assay).

To demonstrate the sensitivity of the StAZ to detect steatosis induction, reference molecules known to induce hepatic lipid accumulation such as ethanol, valproate, amiodarone, and TCDD [24,88,89,90] were first tested; we also evaluated the impact of a high-fat diet, already known to induce steatosis in zebrafish larvae at 5 dpf after one day of feeding [35]. Following these treatments, liver lipid accumulation was investigated using Nile red (NR) staining. As previously described, NR is a selective fluorescent lipophilic dye that exhibits green fluorescence (λ_ex_ 488 nm; λ_em_ 530 nm) when bound to neutral lipids, which enables intracellular lipid droplet detection and quantification of hepatic lipids by high-resolution confocal microscopy imaging [91,92,93]. To normalize the measurement of NR green-fluorescence intensity, specific of lipid, we used the blue (λ_ex_ 405 nm; λ_em_ 488 nm) auto-fluorescence signal. A steatosis score was calculated for each molecule as described in Materials and Methods.

Among the reference treatments tested, only the HFD-fed larvae showed a significant increase in the steatosis score calculated by our test (Figure 2), indicative of lipid accumulation in the liver. For ethanol exposed larvae, despite the large induction of the steatosis score, this was not significant. However, this therefore was in line with steatosis initiation following HFD feeding or ethanol consumption, as previously reported in literature [35,88,94,95]. The other steatogenic molecules failed to induce a significant steatosis with the StAZ, but they showed a trend towards an increase, notably for amiodarone and valproate. These results will be discussed below, and could indicate a low sensitivity of the StAZ test.

Subsequently, the StAZ bioassay was used to screen 10 EDCs belonging to different classes or families of chemicals and chosen for their relevance to human exposure and potential toxicity (Table 1).

We first investigated the toxicity of these EDCs on mortality and larvae morphology to determine non-toxic concentrations of each molecule (Table 2 in Materials and Methods). We then exposed zebrafish larvae to these concentrations ranging from 1 nM (relevant to human exposures or doses found in the environment) to 10 µM (considered a high dose). Screening of these EDCs with the StAZ identified DDE as able to significantly increase the steatosis score at the highest doses of 1 and 10 µM, in a dose-dependent manner. Regarding the other EDCs, even if a trend for paraben, bisphenols, and phthalates was found, there was no significant change of the steatosis score (Figure 3).

### 2.2. Optimization and Validation of the StAZ Bioassay Using a Blue Fluorescent Liver Transgenic Zebrafish Model

In order to optimize the StAZ and to confirm the steatogenic effect of DDE, we used a transgenic line of zebrafish (AB-Tg(Fabp10: CFP; cry: mRFP). In this zebrafish line, the gene of the blue fluorescent protein CFP (cyan fluorescent protein) is under the control of the liver-specific promoter of fabp10 (fatty acid binding protein 10, alias liver fabp); thus, these fishes have a blue fluorescent liver. This model, in addition to the ease of detection of the liver on fluorescent confocal images, also generates a better signal to normalize the liver Nile red fluorescence (the fluorescence intensity is stronger than the autofluorescence used in wild-type animals and the signal is limited to the liver).

Using this model, 48 h treatments with HFD or TCDD (1 nM) were found to induce a significant increase in the steatosis score in contrast to ethanol (Figure 4a). Moreover, we validated the effect of the highest doses of DDE, and observed a significant induction of steatosis both by the increase of steatosis score (Figure 4b) and by the increase in hepatic lipid droplets amount observed by microscopy (Figure 4c). Furthermore, the use of this transgenic zebrafish line appeared to increase the sensitivity of the assay as the scores obtained for DDE were higher.

### 2.3. The Steatogenic Effect of DDE Involves Perturbation of Lipid Metabolism

In order to determine how DDE induced excessive hepatic lipid accumulation, we performed a large scale RT-qPCR gene expression analysis by using the microfluidic Fluidigm-Biomark technology, which allows to carry out 9216 PCR reactions in parallel (96 samples × 96 assays) [96]. In addition to genes involved in lipid homeostasis, genes related to liver toxicity and oxidative stress, and nuclear receptors were investigated because of their involvement in NAFLD. As seen on the heatmap, most of the changes induced by DDE occurred at doses of 1 and 10 µM, which was coherent with the effects observed on steatosis (Figure 5).

Expression of genes involved in lipid homeostasis:

In line with an accumulation of FA in lipid droplets, treatment with DDE at 1 and 10 µM considerably induced genes involved in de novo lipogenesis (DNL), scd1 (Stearoyl-CoA desaturase-1) and me1 (malic enzyme 1), and genes related to FA storage, perilipin 1 and 2 (plin2 and plin3). In addition, expression of genes related to FA uptake were reduced following DDE exposure (fabp10a, fabp11b and cd36; Figure 6a). We also observed a non-significant decrease in the expression of cpt1aa and apob, which are markers of fatty acid β-oxidation in mitochondria and fatty acid export, respectively. In contrast, expression of dgat1 (diacylglycerol O-acyltransferase 1), known to catalyse the incorporation of diacylglycerol in triglycerides for their storage in lipid droplets, was reduced by DDE.

Expression of genes related to sphingolipid, phospholipid, and cholesterol metabolism:

To get further insight into the pathways involved in the development of steatosis induced by DDE, we next investigated the expression of genes involved in the metabolism of sphingolipid and cholesterol in response to DDE exposure. A decrease in cholesterol synthesis was suggested with a down-regulation of hmgcra (hmg-coa reductase) and dhcr7 (7 dehydrocholesterol reductase), involved, respectively, in the first and the last step of cholesterol biosynthesis. An increase in sphingolipid synthesis was suggested, with an up-regulation in the expression of smpd1 (sphingomyelin phosphodiesterase 1) and smpdl3a (Sphingomyelin Phosphodiesterase Acid Like 3A), coding for sphingomyelinase, involved in ceramide synthesis, one of the most common sphingolipids. Our analysis further revealed significant down-regulation of cerS1 (ceramide synthase S1) expression, involved in ceramide synthesis (Figure 6b). Overall, these results suggest that DDE can perturb cholesterol and sphingolipid metabolism.

Expression of nuclear receptors and other transcription factors involved in the regulation of lipid homeostasis:

Most of the effects of EDCs are due to their interactions with nuclear receptors, also known as metabolic receptors. As shown in Figure 6c, the mRNA expression of the nuclear receptors pxr and ppara (peroxisome proliferator activated receptor α), described to be involved in hepatic lipogenesis and β-oxidation, were significantly induced upon DDE exposure.

Concomitantly, we found that DDE reduced the expression of several genes also involved in the regulation of fatty acid metabolism such as pparg (peroxisome proliferators activated receptor γ), rxraa (retinoic x receptor α a), and srebf1 (Figure 6c).

Expression of gene markers of hepatic toxicity:

Exposure to DDE, at the highest doses of 1 and 10 µM, increased the expression of toxicity and inflammation genes. As shown in Figure 6d, DDE up-regulated the expression of il1b (interleukin 1β) and down-regulated that of crp (C-reactive protein), tfa (transferrin α), and cat (catalase), indicating inflammation and liver damage caused by DDE exposure. In addition, the highest doses of DDE caused up-regulation of the anti-oxidant genes hmox1 (heme oxygenase 1) and nqo1 (NAD(P)H Quinone Dehydrogenase 1), reflecting a potential induction of oxidative stress (Figure 6d).

Comparison between HFD and DDE effects on gene expression:

Looking to responses between HFD and DDE, there are only three genes showing same kind of variation (me1, plin3 and srebf1); looking at scd1 expression, note that opposite effects of these two treatments were detected (Figure 6).

### 2.4. Assessment of the Mitochondrial Dysfunction Mediated vy DDE Exposure in Zebrafish Larva

Due to the well-recognized role of mitochondrial dysfunction in NAFLD [97], and based upon our gene expression results suggesting both a decrease in β-oxidation and potential oxidative stress, we examined the expression of genes implicated in mitochondria metabolism (Figure 7a). Our results showed only slight changes in mitochondria-related gene expression (genes related to ATP synthase or respiratory chain complex). Nevertheless, a down-regulation could be observed upon DDE 1 µM of sirtuin genes (sirtuin deacetylase 1 and 3), known to be implicated in the regulation of mitochondrial activity and cellular redox control. We also found increased expression of the pparγ coactivator-1α (ppargc1α), an inducible transcriptional coactivator involved in the regulation of energy metabolism and mitochondrial biogenesis. Finally, a down-regulation of abcg2a expression (a known mitochondrial transporter of metabolites) was also detected.

As DDE was found to dysregulate some genes involved in mitochondrial metabolism, we evaluated mitochondrial respiration in DDE-exposed zebrafish larvae. To this end, we used a protocol adapted from Raftery et al. [98] based on the Agilent Seahorse technology, which allows the measurement of oxygen consumption from live zebrafish larvae. Using these experimental conditions, we observed no significant effect of DDE on mitochondrial respiration (Figure 7b). Altogether, our data do not indicate an effect of DDE on oxidative phosphorylation following a 48 h exposure to 0.1, 1, and 10 µM of DDE.

### 2.5. DDE-Induced Membrane Remodeling in Zebrafish Larvae

One of the main processes identified through our gene expression analysis was a perturbation of the metabolism of sphingolipid and cholesterol, main constituents of the plasma membrane. Biochemical alteration of plasma membrane lipid content is one of the features of plasma membrane remodeling which is linked to alterations of plasma membrane fluidity or modifications of lipid raft properties. Moreover, membrane remodeling has already been shown in the context of steatosis, and our team previously identified it as a key mechanism in the progression of steatosis towards a steatohepatitis-like state in zebrafish larvae co-exposed to B[a]P and ethanol [99]. Therefore, we investigated if DDE could induce such membrane remodeling by analysing membrane order with the fluorescent hydrophobic probe—di-4-ANEPPDHQ. This allowed us to calculate a generalized polarization (GP) value representative of the membrane order that depends on the chemical and physical properties of membranes—lipid composition and packing, fluidity and lipid bilayer thickness, and local hydration.

After staining whole zebrafish larvae with di-4-ANEPPDHQ, liver images were acquired by computing the GP value obtained from fluorescence images of lipid bilayers with low-membrane lipid order—the liquid disordered (Ld) phase, and with high-membrane lipid order—the liquid ordered (Lo) phase, allowing us to estimate membrane order in the liver, as previously described [99]. Exposure of zebrafish larvae to DDE induced a global dose-dependent decrease of membrane order in liver cells, with a significant effect being observed at the highest concentration (Figure 8). As the GP factor depends on the degree of rigidity of the membrane, DDE exposure appears to induce membrane remodeling, with a fluidization of liver membranes, which is coherent with the suggested inhibition of cholesterol synthesis and scd1 increase (Figure 6a,b). These data suggest that membrane remodeling is involved in the steatosis induced by DDE in zebrafish larvae.

### 2.6. Involvement of scd1 in the Pro-Steatogenic and Membrane Remodeling Effects of DDE

Our data indicate that the pro-steatogenic effect of DDE is probably mediated by an increase in de novo lipogenesis and specifically by scd1 up-regulation. In order to firmly confirm that scd1 is involved in the initiation of steatosis induced by DDE, zebrafish larvae exposed to DDE were treated with an inhibitor of SCD1 activity (A939572). As illustrated in Figure 9, the SCD1 inhibitor at 5 µM significantly reduced DDE-induced lipid droplet accumulation in the liver of zebrafish even though a significant increase of steatosis score was still observed with 10 µM DDE in the presence of the inhibitor.

SCD1 catalyses the synthesis of mono-unsaturated fatty acids (MUFA) from saturated fatty acids (SFA); so, it is a major regulator of membrane fluidity by controlling the MUFA/SFA ratio [100]. Because of the involvement of SCD1 in the initiation of steatosis and its role in the modification of the MUFA/SFA ratio, we investigated whether SCD1 was involved in DDE-induced membrane remodeling. We found that the SCD1 inhibitor, A939572 (5 µM), largely reduced DDE-induced membrane fluidization, thus demonstrating the involvement of SCD1 in DDE-induced membrane remodeling (Figure 9b).

## 3. Discussion

The main objectives of this study were to develop an in vivo bioassay in zebrafish larvae to evaluate the capacity of chemicals, especially EDCs, to induce liver steatosis, and then to use this test to screen a panel of 10 EDCs. This allowed us to identify DDE as a highly steatogenic molecule and we further investigated the potential mechanisms involved.

We chose to develop our assay with the zebrafish larva model as it combines the advantages of in vivo models (integration of cell and organ interactions) while being considered an alternative to animal experimentation [42]. In order to evaluate lipid accumulation in the liver, we used an imaging approach based on staining with the lipophilic fluorescent dye, Nile red. To develop and challenge our assay, we used a set of chemicals or diet classically described to induce steatosis in rodent or human as well as in similar zebrafish models. However, in our model, some of these chemicals failed to induce a significant increase in the steatosis score generated by our test (Figure 2b). In fact, in our experimental conditions, amiodarone and valproate were found to be highly toxic in comparison to what was reported in the literature. We were unable to reach concentration levels reported to cause steatosis (10 µM for amiodarone or from 200 to 600 µM for valproate) without excessive mortality (Appendix A).

Nevertheless, steatosis could be induced with a high-fat diet, in line with previous works [35,88,94,95]. In this context, we decided to use our test to screen a panel of 10 selected EDCs (Figure 3). Among these, only one compound, DDE, was found to significantly increase the steatosis score and its impact was even more pronounced than that of the HFD. Of the other tested EDCs, none induced significant increases in steatosis in our assay, although they have been linked to the development of steatosis in the literature. It should be note that these pro-steatogenic effects were not all described in zebrafish larva and not all through direct assessment of liver lipid content. For example, BPA was described to increase liver lipid content associated with an up-regulation of hepatic SREBP and de novo lipogenesis-related genes in rodent and zebrafish adult models [101,102]. However, in the same experimental conditions, BPA at a concentration of 1 µM failed to induce significant hepatic lipid accumulation in zebrafish larvae despite a similar trend, as we observed in our model [49]. Among bisphenols, exposure to BPS has been shown to induce lipid visceral accumulation, although quantification of hepatic TG is lacking [44]. To our knowledge, only the long-term effects of BPS on the development of hepatic steatosis have been performed and only on adult zebrafish [48,103]. DEHP was described to promote lipid accumulation through PPARα/SREBP1c signalling based on in vitro studies [104]. Even though low doses of DEHP (5.8 nM) were shown to modulate the expression of liver genes related to FA metabolism [61], only exposure to the main DEHP metabolite, MEHP at 200 µg/L, was shown to increase liver lipid accumulation in zebrafish larvae at 120 hpf [105]. Many studies have also linked PFOS to the induction of steatosis in adult and larval zebrafish, through a decrease in mitochondrial β-oxidation, an increase in FA synthesis, or an induction of endoplasmic reticulum stress [33,70,71]. In addition, previous studies on zebrafish exposed to cadmium have shown an accumulation of hepatic TG and LD associated with increased lipogenesis and induction of oxidative stress but only in adult zebrafish and at higher concentrations than the ones tested in this study [83,84]. Overall, our results are consistent with the observations from the literature and the discrepancies can be attributed to differences of exposure doses or stages of zebrafish. This lends support to the reliability of our assay.

A reason that could explain the lack of positive readout in our assay of certain compounds would be the limited exposure time of 48 h; increasing exposure time or concentrations could then accentuate lipid accumulation. However, our aim was to use zebrafish larvae as an alternative model to animal experimentation, i.e., before the age of 5 dpf. In addition, as the liver appears to be functional at 3 dpf, it seemed reasonable to keep an exposure window of 48 h (3 to 5 dpf).

Because of the relatively low sensitivity of our zebrafish larva assay, we thought to optimize the test by using a transgenic zebrafish line engineered to have a fluorescent liver. This model facilitates the acquisition of confocal images, as the liver is easily detected. In addition, the stronger blue liver fluorescence in the transgenic zebrafish improved the normalization of the Nile red fluorescence compared to that performed based on the autofluorescence blue signal observed in the wild-type zebrafish. We tested this optimized method with positive control compounds and with DDE (Figure 4) and it globally gave better results; the steatosis score for DDE was even greater. One criticism that could be raised is that the liver blue reporter protein is expressed under the control of the fabp10a gene promoter, and thus, if its expression is decreased, it could give an artefactual increase in the steatosis score. This was actually the case in response to DDE exposure causing both a decrease in fabp10a mRNA levels (Figure 6a) and in the intensity of the blue fluorescent signal. Hence, this might explain why the steatosis score was higher for DDE in the Liblue model than in the wild-type zebrafish model. However, fluorescence of the CFP signal could also be considered as a marker of hepatotoxicity [88] and thus, it could also indicate that DDE, at 1 and 10 µM, is deleterious to the liver. In this context, the CFP signal might reflect liver integrity (quantity of hepatocytes, transcriptional and metabolic functions). Therefore, if a liver with a loss of integrity (marked by a decrease of CFP intensity) is loaded with a high quantity of lipid (higher Nile red staining), it seems reasonable to think that the chemical responsible for these effects could be both hepatotoxic and steatogenic. Taken together, the use of the Liblue model facilitates screening and, in the future, a specific analysis of the CFP signal could be integrated in the analytical processes of our test to evaluate liver toxicity at the same time as steatogenic chemical properties.

Overall, the screening of the 10 EDCs selected using our test allows us to identify one of them, DDE, as an inducer of steatosis. However, improvements are still needed to improve sensitivity. In addition, our test could also be used to screen molecules able to reduce steatosis, for example, in co-treatment with HFD. Indeed, a quite similar approach has been recently successfully developed for small-scale screening of protective molecules towards hepatic liver accumulation on zebrafish larvae between 7 to 8 dpf [104].

To get insight into the underlying mechanisms of DDE’s steatogenic properties, the expression of a set of 89 genes (Figure 5 and Figure 6) was analyzed upon DDE exposure. In coherence with results on steatosis, the built heatmap (Figure 5) clearly showed an effect of DDE exposure at the concentrations of 1 and 10 µM. Focusing on the genes related to fatty acid metabolism (Figure 6a), the steatogenic effects of DDE were consistent with the up-regulation of perilipin 1 and 3, which are key structural proteins of lipid droplets [106]. Regarding the potential mechanisms of the accumulation of fatty acids into lipid droplets, the strongest positive impact of DDE exposure was on de novo lipogenesis-related genes, i.e., me1 and scd1 (Figure 6a). Malic enzyme me1 is particularly interesting, given that this enzyme is involved in fatty acid and cholesterol biosynthesis by generating a required co-factor, NADPH [107]. In addition, an association between increased ME1 gene expression and liver lipid accumulation has already been observed in different steatosis models, notably following high-fructose diet in mice [107,108]. Thus, we could hypothesize that the induction of me1 by DDE participates in the liver lipid accumulation. This remains to be tested. Some genes usually involved in steatosis have been described to have decreased expression in our study. For example, the expression of genes encoding FA transporters (fabp10a, fabp11b and cd36) are decreased in our study, which may be paradoxical but considered as a compensatory mechanism to limit the entry of FA into the cell and prevent their accumulation. Similarly, the expression of dgat1, known to catalyse the final step of TG synthesis for storage in lipid droplets, is reduced by DDE. However, several isoforms of dgats exist and it is worth noting that compensation might exist between the different isoforms of dgats [109].

Changes in gene expression induced by HFD and DDE have been shown to considerably differ (Figure 6). In fact, as molecular pathways differ (increase of lipid uptake, decrease of lipid export, decrease of lipid oxidation…) depending on the cause of steatosis, it is thus not surprising to observe such different profiles in gene expressions. Indeed, this is one of the reasons why we developed this test, i.e., to directly measure the accumulation of lipids, that is, the physiological characteristic of steatosis, independently of the pathways leading to this accumulation, rather than studying this indirectly by one very specific pathway.

Regarding the up-regulation of scd1, it has previously been shown in other models that it can lead to NAFLD [110], especially as it is required for the synthesis of hepatic triglycerides and cholesterol esters [111]. SCD1 is an endoplasmic reticulum-bound microsomal enzyme that catalyses the formation of monounsaturated fatty acids (MUFA) from saturated fatty acids (SFA), by incorporating a double bond at the delta-9 position. Its substrates are palmitic acid (C16:0) and stearic acid (C18:0) and they are converted, respectively, by SCD1 to two MUFA, palmitoleic (C16:1) and oleic (C18:1) acids. Both are major elements for the synthesis of triglycerides and cholesterol esters (principal components of lipid droplets), and of phospholipids [110,111]. Thus, SCD1, a rate-limiting enzyme of these processes, is also a major regulator of membrane fluidity by controlling MUFA/SFA ratios [100].

In this study, we demonstrated that scd1 is a key enzyme involved in the steatogenic effect of DDE as the increase in lipid accumulation by DDE was significantly reduced by scd1 inhibition (Figure 9a). To our knowledge, this is the first time that such a mechanism has been observed concerning the steatosis induced by DDE. A large number of nutritional and hormonal factors finely regulate SCD1 expression through several transcriptional factors like SREBP-1c, LXR, PPAR-a, C/EBP-a, NF-1, NF-Y, AP-1, Sp1, TR, and PGC1-a [112]. In addition, it has also been shown that PXR can induce SCD1 expression directly or indirectly through LXR and SREBP1 [113,114]. DDE has been reported to activate PXR [115], and accordingly, we found an increase in PXR expression following DDE exposure (Figure 6c). In this context, we could hypothesize that PXR activation by DDE would be responsible for scd1 induction in our model.

In the present study, DDE exposure significantly induced zebrafish liver cell membrane remodeling by decreasing the overall membrane order that reflects increased membrane fluidity. A link between membrane remodeling and steatosis has already been demonstrated [116,117,118], in particular with a spatial redistribution of membrane phospholipids in the liver. In fact, effects on the composition of phospholipids ultimately determine membrane fluidity; for this reason, the role of SCD1 is considerable in the physiology of cell membranes. The degree of fatty acid unsaturation in membrane phospholipids affects many membrane-associated functions and can be influenced by altered activities of lipid-metabolizing enzymes such as fatty acid desaturases like SCD1. Moreover, SCD1 expression affects the fatty acid composition of membrane phospholipids, triglycerides, and cholesterol esters, and inhibition of scd1 activity has already been shown to increase membrane saturation [111]. Increased desaturation of newly synthesized fatty acids increases substrate availability for synthesis of unsaturated phospholipids. Furthermore, the generated fatty acyl chains of polyunsaturated phospholipids assume a flexible and bent conformation, resulting in decreased membrane order. This might explain why inhibiting scd1 activation significantly hampered the membrane fluidization upon DDE, as observed in Figure 8b. Scd1 up-regulation and potential consequent changes in MUFA/SFA ratios are probably not the only mechanisms involved. Indeed, reversion of DDE-induced membrane remodeling by scd1 inhibitor was only partial at 5 µM (Figure 8b) and ineffective at lower concentrations able to reduce steatosis. Inhibition of cholesterol synthesis, suggested by our gene expression analyses, could also be involved, as well as changes in ceramide and sphingolipid metabolism (Figure 6b). Further studies should be performed to determine the effect of DDE exposure on the lipid composition in the liver and particularly in hepatocyte membranes to better understand to which extent each process is involved. Finally, as membrane remodeling has been described in the pathogenesis of NAFLD [116,117,118], it could be interesting to precisely determine its role in the context of DDE exposure both following acute (between 3 to 5 dpf) or long-term exposures in steatosis and in its progression to NASH using a known modulator of membrane remodeling. Indeed, we previously demonstrated that membrane remodeling was involved in the progression of steatosis towards steatohepatitis upon BaP/ethanol co-exposure [99]. Therefore, it could also be interesting to evaluate other EDCs for their effect on membrane remodeling and NASH development even if they did not present steatogenic properties in our conditions. Still, in the context of metabolic diseases, it could also be interesting to extend investigations on membrane remodeling upon DDE exposure to other tissues such as pancreas, gut, or adipose tissue, and to determine if DDE induced-membrane remodeling is specific to the liver or not.

## 4. Materials and Methods

### 4.1. Zebrafish Maintenance

Zebrafish (wild-type AB strain and the Liblue transgenic lines (AB-Tg(Fab10: CFP; cry: mRFP), ZeClinics Headquarters, Barcelona, Spain)) were handled, treated, and killed in agreement with European Union regulations concerning the use and protection of experimental animals (Directive 2010/63/EU). Fertilized zebrafish embryos—collected following natural spawning—were obtained from Biosit (Rennes, France). Embryos and larvae (sex not yet established at this stage) were maintained in bath water (90 µg/mL Instant Ocean [Aquarium Systems, Sarrebourg, France], 0.58 mM CaSO_4_, 416 µg/L methylene blue, dissolved in reverse-osmosis purified water) at 29 °C with an 14L:10D photoperiod under static conditions.

### 4.2. Chemicals and Larva Exposure

Bisphenol A (BPA, Cat No 239658), Bisphenol S (BPS, Cat No 103039), Bisphenol F (BPF, Cat No B47006), Di(2-ethylhexyl) phtalate (DEHP, Cat No 36735), dibutylphtalate (DBP, 524980), Perfluorooctanesulfonic acid, (PFOS, Cat No 77282), Perfluorooctanoic acid (PFOA, Cat No 171468), Dichlorodiphenyldichloroethylen (DDE, Cat No 35487) Cadmium (CdCl_2_, Cat No 202908), butyl-paraben (BtP, Cat No 54680), Valproic acid (Cat No P6273), Amiodarone hydrochloride (Cat No A8423), SCD1 inhibitor (A939572; SML2356) were all purchased from Sigma-Aldrich (St. Louis, MO, USA). 2,3,7,8-Tetrachlorodibenzo-p-dioxin (TCDD;CIL-ED-901) was purchased from LGC Standard. Of each compound, 1000× stock solutions were prepared in dimethyl sulfoxide (DMSO) and stored at −20 °C; final dilutions of 0.1% *v*/*v* were used.

The exposure protocol is shown in Figure 1. Briefly, at 3 days post-fertilization (dpf), zebrafish eleuthero-embryo (simply called larvae hereinafter) were selected for the absence of morphological defects (edema, spinal curvature, impaired swimming). Selected larvae were then incubated in glass containers at a density of 1 larva/ mL of medium, containing endocrine disruptors (0.001, 0.01, 0.1, 1 or 10 µM) or vehicle alone (0.1% DMSO) directly added to the incubation medium. The toxicities of each EDC were determined upfront and concentrations used generally ranged from levels found in population (see Table 1, concentration in the nM range) to concentrations below the threshold inducing larvae mortality (Table 2). Exposures were maintained from 3 to 5 dpf. At the end of the exposure period, 5 dpf-old zebrafish were processed for fluorescence image analysis to evaluate liver lipid content.

As positive controls, zebrafish were fed daily during 1 h before medium renewal with a high-fat diet (HFD) (HFD; dried chicken egg yolk containing around 53% fat; Sigma-Aldrich St. Louis, MO, USA). Validation of the protocol to induce liver steatosis has been previously published [35,36]; of note, lipid accumulation was clearly detected as soon as 1 day of HFD.

### 4.3. Neutral Lipid Staining with Nile Red

At 5 dpf, after 48 h exposure, zebrafish larvae were washed in phosphate buffered saline (PBS) and then fixed in 4% paraformaldehyde in PBS at 4 °C. A staining protocol of liver neutral lipids using Nile red was adapted from previous reports [119,120]. After washing in PBS, whole larvae were stained for 1 h with Nile red at 5 µg/mL (N3013, Sigma-Aldrich; stock solution was prepared at 100 µg/mL in acetone). The larvae were then washed twice in PBS and placed in baths of increasing glycerol concentrations (20–50–80%), before being mounted on slides with PBS 20%/glycerol 80%. Images of zebrafish larvae were acquired as described below.

### 4.4. Quantitative Analysis of Hepatic Lipid Fluorescence Signals

To evaluate neutral lipid content, fluorescence images of zebrafish larvae stained with Nile red were acquired with a confocal fluorescence microscope LEICA TCS SP8 (Leica Microsystems, Wetzlar, Germany). A first image—characteristic of neutral lipid fluorescence—was taken under excitation at 488 nm using an argon ion laser with a photomultiplier tube (PMT) range of 500–580 nm, whereas a second image—insensitive to neutral lipids—was taken under excitation at 405 nm with a diode laser with a PMT range of 450–480 nm. Using Fiji imaging processing software (ImageJ, National Institutes of Health, Bethesda, MD, USA) with three developed macros (Appendix A), liver area was delimited by user for each larva with the first one, and then, with the two others, three parameters were calculated:(1)Ratio: this is the value corresponding to fluorescence intensity of image A in liver area divided by the fluorescence intensity of image B in the liver area (F_green_/F_blue_). To use it in score calculation, this ratio is corrected and divided by the mean of same fluorescence ratio determined for the group of larvae exposed to DMSO.
$${\mathrm{Ratio}}_{x}=\frac{\left(\;{\left(\frac{F_{\mathrm{green}}}{F_{\mathrm{blue}}}\right)}_{x}-\mathrm{mean}\;{\left(\frac{F_{\mathrm{green}}}{F_{\mathrm{blue}}}\right)}_{\mathrm{DMSO}}\right)}{\mathrm{mean}\;{\left(\frac{F_{\mathrm{green}}}{F_{\mathrm{blue}}}\right)}_{\mathrm{DMSO}}}$$

(2)LD_dens: this is the density of lipid droplets, which corresponds to the amount of lipid droplets reported to the surface of the liver.(3)LD_area: this value is the surface occupied by lipid droplets in liver area.

Concerning the integration of these 3 parameters to generate the steatosis score, even if the density of droplets and the area of droplets could seem more robust than the fluorescence ratio of the Nile red signal to the normalization signal, depending on the nature of steatosis, each one could be important. In fact, liver lipid accumulation could have different phenotypes from microsteatosis to macrosteatosis and thus, the profile of lipid droplets could differ greatly with sizes ranging from few nm to more than 100 µM. As the number of lipid droplets per unit of liver area could not be sufficient to give a proper measurement of steatosis, it was decided to also take into account the area of lipid droplets. However, it seemed that such a procedure was still not convincing since the size of small lipid droplets could be under the limit of microscopic resolution or not sufficiently discriminated by the automatized analytical process. In order to compensate for this, fluorescence ratio taken as a global evaluation of the liver content in neutral lipid was also integrated in steatosis score calculation.

Each parameter was pondered by a specific coefficient. These coefficients were set empirically in order to give a homogenous weight to each parameter in the score and to obtain the highest discrimination with positive and negative control (this is illustrated by the Appendix A).

Based on these three parameters, a steatosis score was calculated as follows.
$$\mathrm{Steatosis}\;\mathrm{score}=\left(\mathrm{Ratio}\times 400\right)+\left({\mathrm{LD}}_{\mathrm{dens}}\times 10000\right)+\left(\mathrm{LD}\_\mathrm{area}\;\times 100\right)$$

To determine a steatosis score for each larva, images of 3 optical slices were taken and analyzed and the mean of these 3 scores was used as the score of this larva.

### 4.5. Analysis of Gene Expression by Fluidigm-Biomark HD System Technology

Gene expression levels in whole zebrafish larvae exposed to EDCs during 48 h (3 to 5 dpf) were assessed using the Fluidigm-BioMark HD system. Briefly, for mRNA extraction, 30 whole zebrafish larvae were pooled and homogenized in 100 μL TRIzol reagent, and total RNA was extracted according to the manufacturer’s protocol with TRIzol reagent (Gibco, Carlsbad, CA, USA). RNA samples were then reverse-transcribed and cDNAs were obtained using the Fluidigm reverse transcription Master Mix, and were then preamplified for 12 cycles in the presence of the Fluidigm Pre-Amp Master Mix. Gene expression was then measured with the Biorad EvaGreen SuperMix on a 96.96 Dynamic Array IFC. After quality control verification, mRNA expression was calculated with the ΔΔCt method and normalized by using actb2, 18 s, and gapdh as reference. Sequences of the tested zebrafish primers and official gene names are provided in Appendix A (information about the primer conditions is available upon request).

### 4.6. Membrane Order Determination by Fluorescence Staining

Plasma membrane order in zebrafish liver was assessed by confocal fluorescence microscopy using the membrane order-sensitive fluorescent probe, di-4-ANEPPDHQ (Molecular Probes, Life Technologies). This probe displays a fluorescent spectral blue-shift from 620 nm, when incorporated into lipid bilayers with a low-membrane lipid order (liquid disordered phase, Ld), to 560 nm, when inserted into lipid bilayers with a high-membrane lipid order (liquid-ordered phase, Lo). After acquisition using confocal fluorescence microscopy of both disordered and ordered-phase fluorescence images, a new image, indicative of membrane lipid order, was obtained by calculating the generalized polarization (GP) value—a ratiometric measurement of fluorescence intensities for each pixel which is associated to membrane lipid order [121]. Briefly, after EDC exposure, larvae were washed in PBS and fixed in 4% paraformaldehyde in PBS at 4 °C. After two washes in PBS, larvae were stained with 10 μM di-4-ANEPPDHQ for 3 h at 28 °C. Larvae were then washed twice in PBS and dipped in several baths of progressively higher glycerol concentrations. After this, they were mounted in 80% glycerol in PBS for observation with a LEICA TCS SP8 confocal fluorescence microscope (Leica Microsystems, Wetzlar, Germany). At 488 nm excitation with an argon ion laser, ordered membrane images were acquired with a photomultiplier tube (PMT) with a range of 500–580 nm, whereas for disordered membrane images, the PMT used had a range of 620–750 nm (magnification 400×). Fiji imaging processing software (ImageJ, National Institutes of Health, Bethesda, USA) and the macro published by Owen et al. [121] were used. GP images were generated according to the following calculation: GP = (I_500–580_ − I_620–750_)/(I_500–580_ +  I_620–750_). In order to avoid potential variations across different batches of larvae or different stainings, for each experiment—one batch of zebrafish larvae/one staining procedure—GP values were expressed as the difference (ΔGP) between individual larva GP value and the mean GP of control larvae (DMSO) within the same experiment.

### 4.7. Assessment of Mitochondrial Oxygen Consumption

To evaluate the oxygen consumption rate (OCR) of mitochondria in zebrafish larvae using the Seahorse XFe24 Analyzer (Agilent Technologies, Santa Clara, CA, USA), we used specific exposure conditions and adapted the protocol from Raftery et al. [98]. Briefly, larvae of 3 dpf were exposed to toxicants for 48 h (3 to 5 dpf). Following treatment, larvae were anesthetized with 31.25 mg/L tricaine (MS-222, Sigma-Aldrich, St. Louis, MO, USA) in bath water. Larvae were then placed in a 24 multi-well plate for Seahorse (1 larva/well). Larvae were fixed at the bottom of the wells with a grid insert, and the volume of bath water was adjusted to 500 µL per well. Twenty min after anesthesia onset, larvae were placed in the Seahorse XFe24 analyzer for assessment of OCR (28 °C, 1 read per cycle of 4 min) using the following phases and inhibitors: Phase 1: 6 cycles (24 min); Phase 2: addition of 2.5 µM FCCP (carbonyl cyanide-p-trifluoromethoxyphenylhydrazone), 8 cycles (32 min); Phase 3: addition of 6.25 mM NaN3 (sodium azide), 20 cycles (80 min). Using Wave software (version 2.6.0, Agilent Technologies), OCR levels were analyzed in order to obtain basal, maximal, and spare mitochondrial and non-mitochondrial respiration levels with at least 4 larvae per condition.

### 4.8. Statistical Analyses

All values were presented as mean ± SEM (standard error of the mean) from at least three independent experiments, except for the StAZ bioassay, for which the number of larvae and independent batches are detailed in the figure legends. Multiple comparisons among groups were performed using one-way analysis of variance (ANOVA) followed by a Dunnett’s post-test/Newman–Keuls post-test, or using a Kruskal–Wallis test followed by a Dunn’s post-test using GraphPad Prism8 software (GraphPad Software, San Diego, CA, USA). Differences were considered significant when *p* < 0.05 (*), *p* < 0.01 (**), *p* < 0.001 (***).

## 5. Conclusions

Overall, in the present study, we have developed an in vivo bioassay that might be useful to screen EDCs potentially implicated in the initiation of steatosis. This steatogenic bioassay, StAZ, enabled us to highlight the steatogenic effect of DDE. This process involves the regulation of lipid metabolism-related genes with an increase in de novo lipogenesis likely dependent on scd1 activation. In addition, a new mode of action of DDE was shown with the involvement of scd1 activation-dependent liver membrane fluidization.

## Figures and Tables

**Figure 1 ijms-24-03942-f001:**
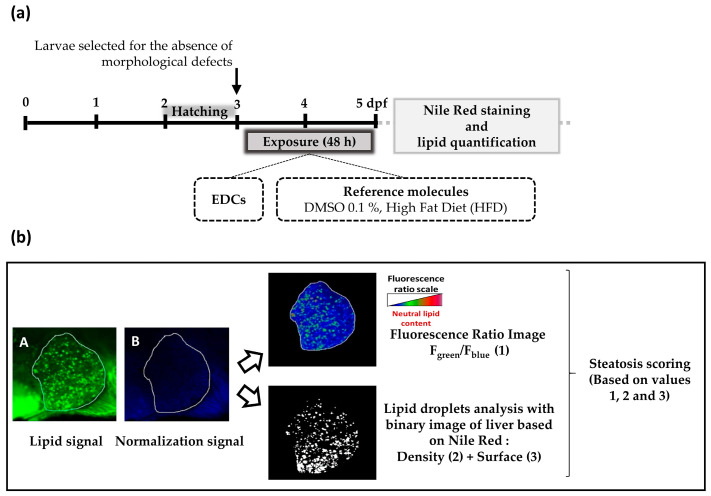
Experimental design of the StAZ bioassay (**a**) Steatogenic Assay on Zebrafish (StAZ) protocol. Zebrafish larvae at 3 days post-fertilization (dpf) were exposed to the selected compounds at non-toxic concentrations, to the positive control (HFD), or to vehicle alone (DMSO 0.1%), during 48 h. Following exposure, larvae were euthanized and fixed, and hepatic lipid accumulation was measured using Nile red staining and fluorescence signal quantification. (**b**) Fluorescent signal quantification workflow. After image acquisition of stained zebrafish larvae with confocal fluorescence microscopy, two types of images were obtained: a first one with a green signal—characteristic of neutral lipid fluorescence, and a second one with a blue signal for normalization—insensitive to neutral lipids (images A and B, respectively). Using Fiji imaging processing software and home-made macros, three parameters were calculated: (1) the ratio of fluorescence intensity of image A to image B per liver area (F green/F blue), (2) the amount of lipid droplets per liver area, and (3) the surface occupied by lipid droplets per liver area. Based on these three parameters, a steatosis score was calculated. Each parameter was pondered by a specific coefficient determined empirically.

**Figure 2 ijms-24-03942-f002:**
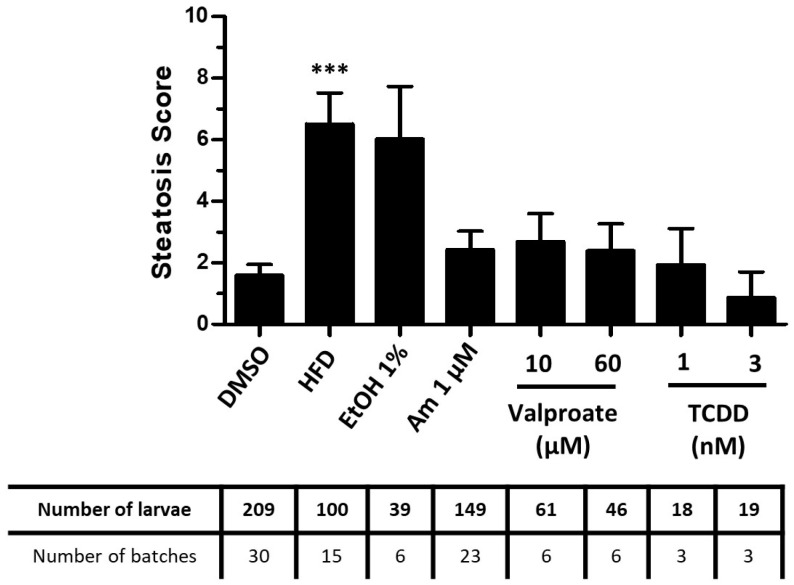
Screening of the steatogenic molecules with the StAZ bioassay. Zebrafish larvae at 3 dpf were exposed to selected steatogenic controls: high-fat diet (HFD), ethanol 1% *v*/*v* (EtOH 1%), amiodarone (1 µM), valproate (10/60 µM) or TCDD (1/3 nM). Following exposure, larvae were euthanized and fixed, and steatosis score (cf. Figure 1b) was calculated based on Nile red staining and image-based automated analysis after confocal microscopy acquisition. Values are mean +/− SEM, with the number of batches and larvae indicated in the table underneath the graph. *** *p* < 0.001, by comparison with control group using Kruskal–Wallis test.

**Figure 3 ijms-24-03942-f003:**
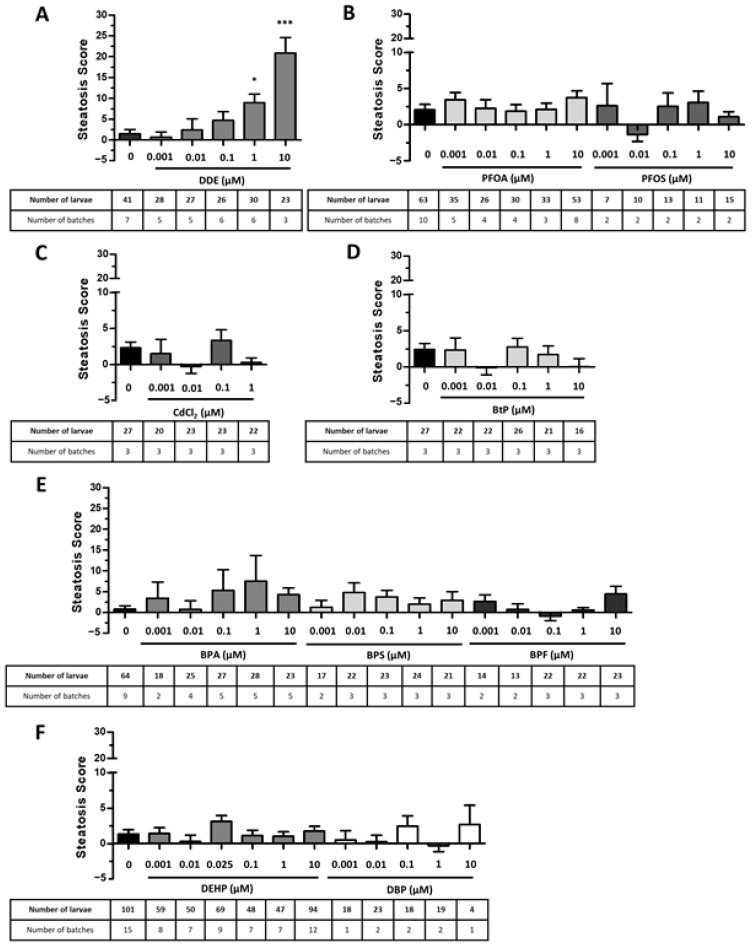
Screening of endocrine disruptors with the Steatogenic Assay on Zebrafish. Zebrafish larvae at 3 days post-fertilization were exposed to the selected compounds at non-toxic concentrations: DDE (**A**), perfluorinated compounds (**B**), cadmium (**C**), butyl-paraben (**D**), bisphenols (**E**), phthalates (**F**) or to vehicle alone, during 48 h. Following exposure, larvae were euthanized and fixed, and steatosis scores (based on lipid fluorescence intensity, lipid droplet density and area in the liver) were calculated based on Nile red staining and image-based automated analysis after confocal microscopy acquisition. Values are mean +/− SEM; with the number of batches and larvae indicated in the table underneath the graph. * *p* < 0.05, *** *p* < 0.001 by comparison with control group using Kruskal–Wallis test.

**Figure 4 ijms-24-03942-f004:**
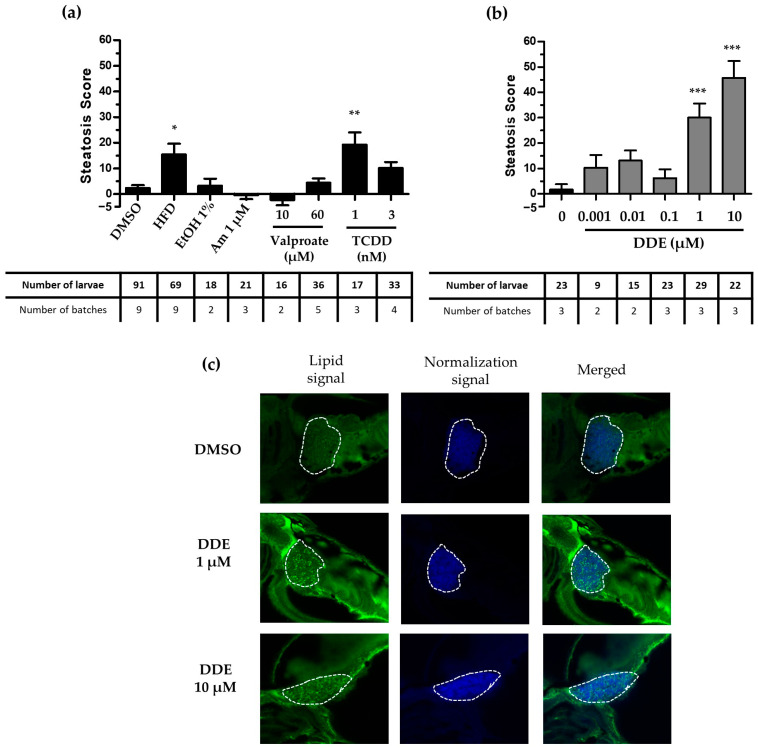
Screening of the steatogenic molecules and validation of the steatogenic effect with the StAZ using the transgenic zebrafish model. Zebrafish transgenic larvae at 3 days post-fertilization were exposed to positive control steatogenic conditions (HFD, ethanol, amiodarone, valproate, TCDD) (**a**), to DDE at concentrations ranging from 0.001 to 10 µM (**b**), or to vehicle alone (**a**,**b**), during 48 h. Following exposure, larvae were euthanized and fixed, and steatosis scores (based on lipid fluorescence intensity, lipid droplet density, and area in the liver) were calculated based on Nile red (NR) staining and image-based automated analysis after confocal microscopy acquisition. Values are mean +/− SEM, with the number of batches and larvae indicated in the table underneath the graph. * *p* < 0.05, ** *p* < 0.01, *** *p* < 0.001, by comparison with control group using Kruskal–Wallis test. (**c**): Confocal microscopy images obtained after DDE (1/10 µM) or vehicle control exposure and Nile red staining showing lipid droplets stained in green. The liver is delimited by the dotted lines.

**Figure 5 ijms-24-03942-f005:**
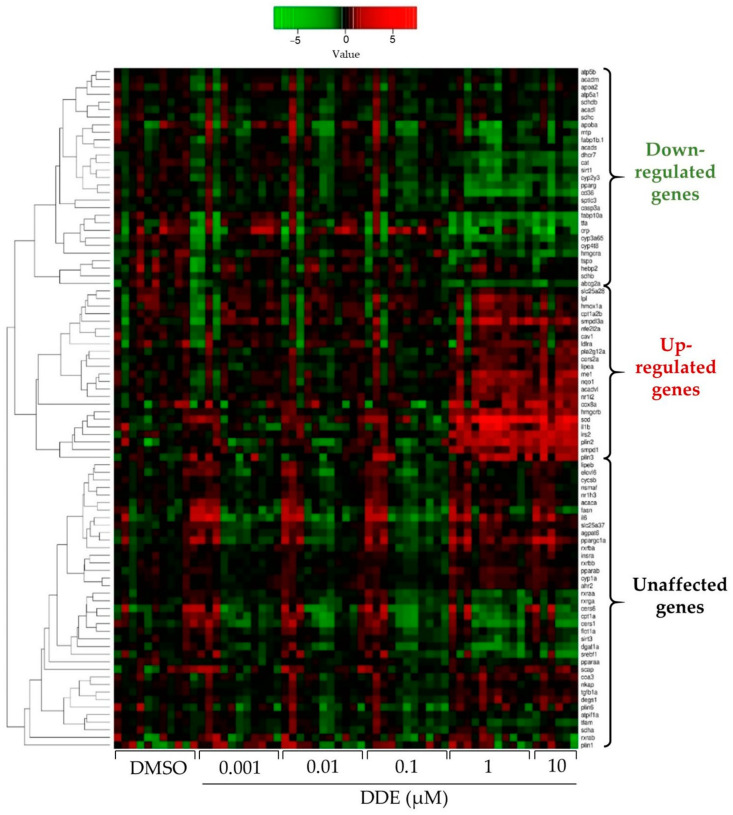
Heatmap showing the impact of DDE on mRNA expression of different genes related to metabolic pathways. mRNA expression was evaluated by RT-qPCR gene expression analysis by using the microfluidic Fluidigm-Biomark technology. Zebrafish larvae were exposed to DDE (from 1 nM to 10 µM) during 48 h, from 3 to 5 dpf. mRNA expressions of 90 genes related to nuclear receptors, lipid homeostasis, liver metabolism, inflammation and oxidative stress were analysed; results are shown as a heatmap (each raw represents an independent mRNA sample extracted from a pool of zebrafish larvae). Data are expressed as mean expressions relative to mRNA levels measured in DMSO control larvae, set at 0 (log 2 fold change) (*n* ≥ 8).

**Figure 6 ijms-24-03942-f006:**
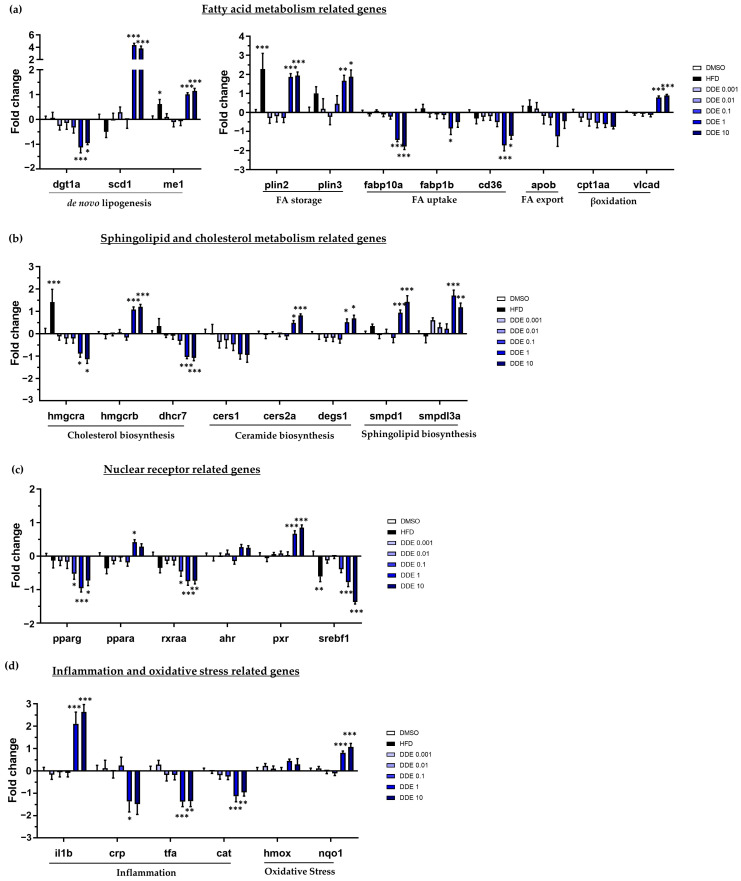
Impact of HFD and DDE on mRNA expression of several genes involved in lipid metabolism characteristic of steatosis and in toxicity. mRNA expression was evaluated by RT-qPCR gene expression analysis by using the microfluidic Fluidigm-Biomark technology. Zebrafish larvae were exposed to HFD or DDE (from 1 nM to 10 µM) during 48 h, from 3 to 5 dpf. mRNA expressions of genes related to fatty acid metabolism (**a**), sphingolipid and cholesterol metabolism (**b**), nuclear receptors, (**c**) and inflammation and oxidative stress (**d**), are shown. Data are expressed relative to mRNA levels measured in DMSO control larvae, set at 0 (log 2 fold change). Values are the mean ± SEM (*n* ≥ 8). * *p* < 0.05, ** *p* < 0.01, *** *p* < 0.001, by comparison with control group using ANOVA and Dunnett’s test.

**Figure 7 ijms-24-03942-f007:**
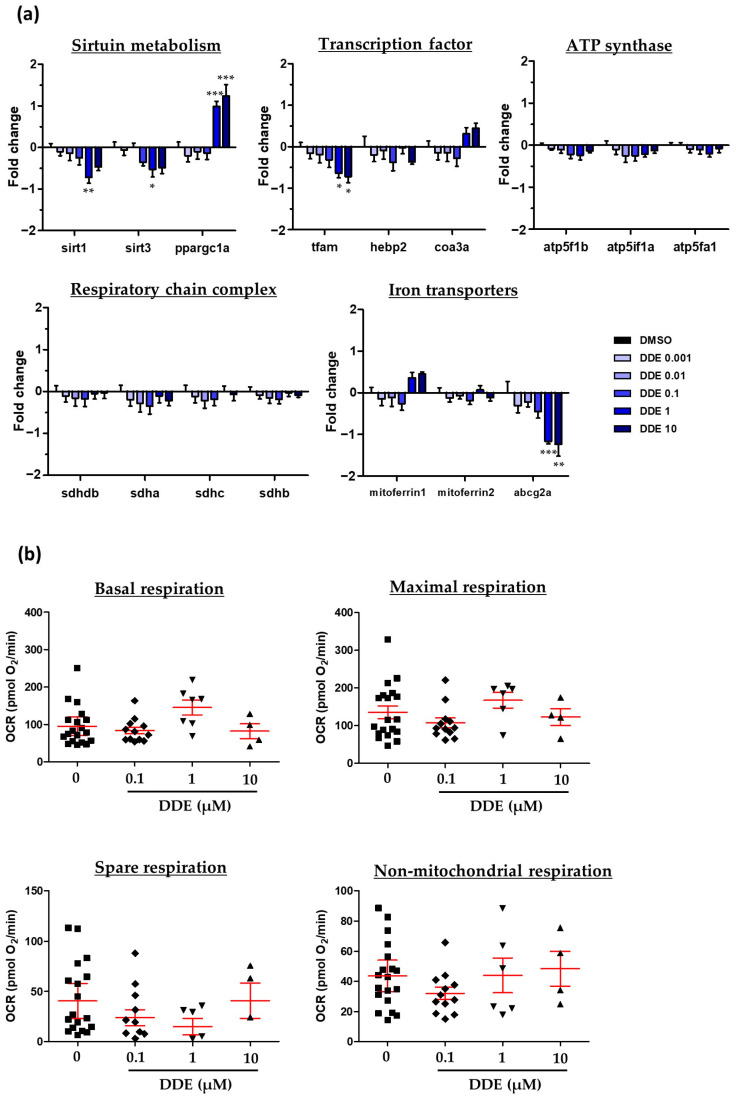
Assessment of potential mitochondrial dysfunction induced by DDE. Zebrafish larvae at 3 dpf were exposed to DDE (from 1 nM to 10 µM) or to control vehicle during 48 h. (**a**) Impact of DDE on the mRNA expression of several genes involved in mitochondrial metabolism. mRNA expression was evaluated by RT-qPCR gene expression analysis by using the microfluidic Fluidigm-Biomark technology. Data are expressed relative to mRNA levels measured in DMSO control larvae, set at 0 (log 2 fold change) (*n* ≥ 8). (**b**) Evaluation of respiration after DDE exposure of zebrafish larvae. Measurements on zebrafish larvae were realized on a Seahorse XFe24 Analyzer of mitochondrial oxygen consumption. Values are the mean of oxygen consumption rate (OCR, in pmol of O_2_/min) ± SEM measured from at least four larvae per condition. Values are the mean ± SEM. * *p* < 0.05, ** *p* < 0.01, *** *p* < 0.001, by comparison with control group using ANOVA and Dunnett’s test.

**Figure 8 ijms-24-03942-f008:**
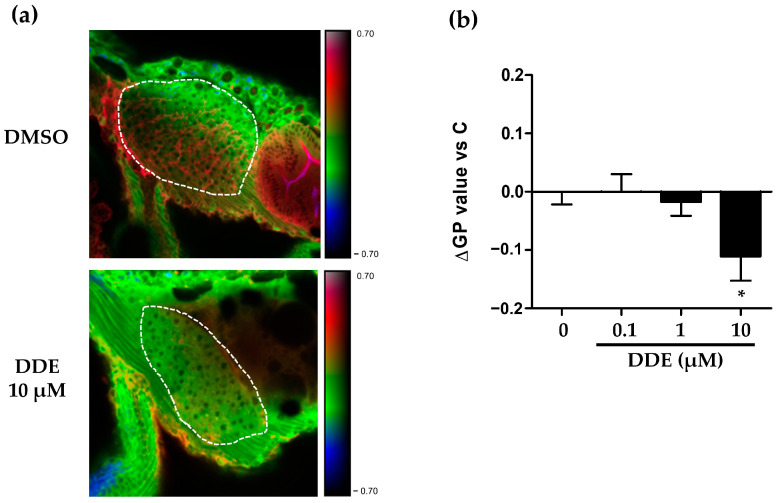
DDE exposure-induced membrane remodeling in the liver of zebrafish larvae. Membrane order characteristic of membrane remodeling was assessed in liver of zebrafish larvae at 3 dpf after exposure for 48 h to DDE (0.1/1/10 µM) or to control vehicle. Zebrafish larvae were stained with di-4-ANEPPDHQ—a membrane order-sensitive fluorescent probe—and analyzed by confocal fluorescence microscopy. Membrane order in membranes of zebrafish liver was measured by computing the generalized polarization (GP) factor. (**a**) Representative liver images of control vehicle (DMSO) and DDE (10 µM) treatment (magnification ×400). The liver is delimited by the dotted lines. (**b**) Changes in GP values (ΔGP) were expressed as the difference between individual larva GP value and the mean GP calculated in control larvae (DMSO). Values are the mean ± standard error of the mean (SEM) of at least 20 larvae. * *p* < 0.05 by comparison with the control group using ANOVA and Dunnett’s test.

**Figure 9 ijms-24-03942-f009:**
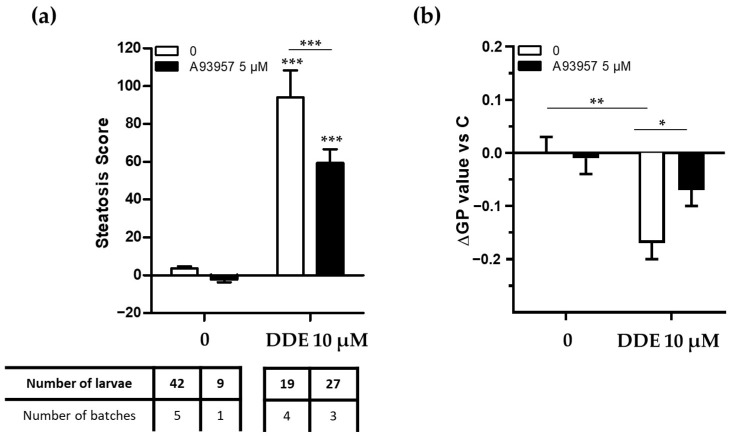
SCD1 inhibition decreases hepatic lipid accumulation and membrane remodeling after exposure to DDE. Zebrafish transgenic larvae at 3 days post-fertilization were co-exposed to a specific SCD1 inhibitor (A939572 5 µM) and to DDE (10 µM) during 48 h. (**a**) For the steatosis score evaluation, larvae were euthanized following exposure and fixed, and steatosis scores (based on lipid fluorescence intensity, lipid droplet density, and area in the liver) were calculated based on Nile red staining and image-based automated analysis after confocal microscopy acquisition. Values are mean +/− SEM; with the number of batches and larvae indicated in the table underneath the graph *** *p* < 0.001, by comparison with control group using two-way ANOVA and Bonferroni tests. (**b**) Membrane order was assessed in liver cells of zebrafish larvae after staining with di-4-ANEPPDHQ—a membrane order-sensitive fluorescent probe—and analyzed by confocal fluorescence microscopy. Membrane order in membranes of zebrafish liver was measured by computing the generalized polarization (GP) factor. Changes in GP values (ΔGP) were expressed as the difference between individual larva GP value and the mean GP measured in control larvae (DMSO). Values are mean +/− SEM for *n* ≥ 3 batches per conditions. * *p* < 0.05, ** *p* < 0.01, by comparison with all groups using ANOVA and Newman–Keuls test.

**Table 1 ijms-24-03942-t001:** List of the EDCs studied in the project and urinary or serum levels detected in the cohort study ESTEBAN.

Category (Main Sources)	Chemical Name Exposure Levels in Human [43]	Metabolic Disruption
**Aromatic compounds, plasticizers** (Food containers, sales receipt, toys)	Bisphenol A (BPA)	**Obesity:** Increases food intake, body weight, and adiposity in vivo [44]; stimulates adipocyte differentiation, lipogenesis; decreases adipocyte insulin sensitivity; activates the cannabinoid system, increases lipogenic gene expression (PPARγ, C/EBPα, LPL, SREBP-1c, FAS, SCD1) in vitro
	0.01–0.45 µM (urine)	
	Bisphenol F (BPF)	**Diabetes:** Increases glucose intolerance and insulin resistance; disrupts pancreatic β cells’ function via mitochondrial disruption and an increase in ROS production [45]
	0.001–0.17 µM (urine)	
	Bisphenol S (BPS)	**NAFLD:** Increases in liver lipid synthesis and accumulation [46,47]; interacts with metabolic receptors (PPARγ) [47,48,49] and lipogenic gene targets (SREBP-1c, FASN, CD36) [18,47,48,49,50]; induces mitochondrial dysfunction [49,51,52] and oxidative stress [51,53,54,55]; induces hepatic inflammation and endoplasmic reticulum stress promoting the progression to NASH [48]
	0.001–1.3 µM (urine)	
**Organic compounds, plasticizers** (Food containers, kitchen utensils, toys, cosmetics, perfumes, medical tubing, adhesives, paints)	Di(2-ethylhexyl) phthalate (DEHP)	**Obesity:** Increases food intake, body weight, and visceral white adipose tissue in vivo; promotes adipocyte proliferation; increases insulin resistance and glucose intolerance in vitro [56]
	45–69 nM (urine)	**Diabetes:** Decreases β-cells’ insulin signalling; induces oxidative stress [57]
	Dibutyl phthalate (DBP)	**NAFLD:** Disrupts liver lipid metabolism through PPARs and/or SREBP1c signalling pathway [58], leading to hyperlipidaemia; induces oxidative stress and inflammation [59,60,61,62]
	62–71 nM (urine)	
**Fluorinated tensoactives** (Stain-resistant coating in clothing, kitchen utensils, home furnishings, food packaging, paint)	Perfluorooctanoic acid (PFOA)	**Obesity:** Induces adipocyte differentiation through PPARγ activation; increases expression of adipogenic genes in vitro and in vivo
	3.7–5.0 nM (urine)	**Diabetes:** Increases insulin resistance and glucose intolerance [63]
	Perfluorooctanesulfonic acid (PFOS)	**NAFLD:** Increases triglyceride (TG) levels and hepatic lipid droplet accumulation [64]; alters lipid metabolism; activates PPARα signalling [65,66,67]; decreases β-oxidation and lipid transport (PFOS) [68,69,70,71] and mitochondrial dysfunction (PFOA) [72]; induces inflammation [73]
	4.4–8.1 nM (urine)	
**Insecticide** (Decomposition product of DDT, insecticides, and repellents (anti-mosquito), food)	Dichloro diphenyl-dichloro ethylene (DDE)	**Obesity:** Increases body weight and fat mass, reduces energy expenditure and thermogenesis in vivo; increases adipocyte differentiation and proliferation; up-regulates adipogenesis mediators (C/EBPα, SREBP1, PPARγ); increases fatty acid uptake and adipokine release (adiponectin, leptin, resistin) in vitro [74,75]
	0.32–0.57 nM (serum)	**NAFLD:** Increases liver lipids level [74] and alters their metabolism [76]; induces liver mitochondrial dysfunction [77,78,79] and oxidative stress in vivo [80]
**Heavy metal** (by-products of waste treatment, food industry, fertilizers, electrical equipments, pigments for paints)	Cadmium (CdCl_2_)	**Obesity:** Increases adiposity [81]
	2.5–13.6 nM (urine)	**Diabetes:** Impairs glucose metabolism leading to insulin resistance, ROS production, mitochondrial dysfunction, and apoptosis of β-cells [82].
		**NAFLD:** Increases lipid accumulation through hepatic lipogenesis [81,83]; promotes oxidative stress [83,84]; causes mitochondrial dysfunction [84]; decreases β-oxidation [85]; dysregulates autophagy [84,86]; increases inflammation [81,83]
**Anti-microbial conservator** (Preservatives in foods, personal care products, and cosmetics)	Butyl-paraben (BtP)	**Obesity:** Promotes adipogenesis in vitro, increases body weight and adipocyte size in vivo [87]
	2.6 nM (urine)	

**Table 2 ijms-24-03942-t002:** Evaluation of EDC toxicity on zebrafish larvae. The toxicity of each compound was evaluated in terms of mortality and morphological damage: green 0–10%, yellow 10–30%, orange 30–80%, red 100% of affected larvae. The dotted lines indicate the concentration ranges of each compound tested with the StAZ.

Mortality								Morphological Damage							
EDC (µM)	0.001	0.01	0.1	1	10	50	100	EDC (µM)	0.001	0.01	0.1	1	10	50	100
BPA								BPA							
BPS								BPS							
BPF								BPF							
DEHP								DEHP							
DBP								DBP							
PFOA								PFOA							
PFOS								PFOS							
DDE								DDE							
CdCl_2_								CdCl_2_							
BtPB								BtPB							

## Data Availability

Not applicable.

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
