# Peer review of "A New In Vivo Zebrafish Bioassay Evaluating Liver Steatosis Identifies DDE as a Steatogenic Endocrine Disruptor, Partly through SCD1 Regulation"

_ijms, 2023, doi:10.3390/ijms24043942_

Round 1

Reviewer 1 Report

In this manuscript, the authors describe an approach to screening for environmental toxicants that induce fatty liver disease using the zebrafish. The rationale behind the screening approach is sound, but the StAZ assay appears to lack  sensitivity, making these results somewhat preliminary. Several other major and minor issues would need to be addressed before publication.

Major issues:

(1)   Only the HFD, ethanol, and one of the EDCs, DDE, caused detectable changes in hepatic lipid droplets using this method, suggesting that the sensitivity and therefore the utility of the screening method are limited. Furthermore, as some of the EDCs tested are expected to result in increased lipid accumulation and/or sdc1 expression, (as highlighted in Table 1), did not meet the threshold of the assay. These nonetheless should be tested in using the RT-PCR expression panel, which is likely much more sensitive than the StAZ screen. This would provide some additional information as to the utility of the StAZ approach. In addition, in the RT-PCR panel, as with the StAZ screen, the results with DDE should be compared to additional NAFLD-inducing positive controls, such as HFD and ethanol.

(2)   Although the scoring algorithm for StAZ is provided in the methods, justification for the weighting of the components of the StAZ score must be provided. Rather, it is simply stated that this was empirically determined. While the density of droplets and the area of droplets are robust measurements, likely to be consistent at all confocal imaging depths, the intensity ratio measurements using two wavelengths of light, one for Nile red, and the other for autofluorescence or CFP normalization, will vary with depth. That is to say, the shorter wavelengths will quickly lose intensity with depth relative to longer wavelengths. For these reasons, each of the three measurements comprising the steatosis score could and should be presented independently, at least as supplemental data.

(3)   It is not clear how many confocal slices were examined per animal, and whether there was any biological or experimental variation with depth or anatomical region of the liver.

(4)   In the discussion, the authors recognize the potential hepatotoxic effects of compounds used in the screen, but did not assess toxicity, which could temper the interpretation of the results. A basic toxicity/viability assay should be performed for the effective doses of the DDE.

(5)   Membrane remodeling changes are one of the most interesting parts of this manuscript, but again, the changes need to be compared to other inducers of NAFLD and other tissues should be examined as well. These tissues might include pancreas and gut regions.

 Minor issues:

(1)   The introduction is much longer than necessary. Could be streamlined by 30-50%

(2)   The authors should provide justification in the methods or results for the EDC concentrations examined with regard to likely levels encountered in the environment.

(3)   A recent study describing a zebrafish screening approach with complementary but opposite objectives—to find small molecules preventing or alleviating hepatic liver accumulation should be referenced (PMID: 35203687).

(4)   Most of the discussion of genes found to have altered expression in the results is unnecessary and at best should be moved to the discussion. Since the underlying speculations on what the changes mean mechanistically are arbitrary hand-waving, the discussion of anything beyond just which genes are changed and what pathways they are part of, is pretty meaningless.

Author Response

We are very grateful to the two reviewers for their commentaries regarding our submitted article and their suggestions for improving it. We have followed their recommendations and have made changes that are described in the attached pdf file in point by point answer to each reviewer.

Reviewer 2 Report

The authors present a novel assay for evaluating liver steatosis in zebrafish, in addition to detailing changes (gene expression and membrane remodeling associated with DDE induced-steatosis. The manuscript is well written, concise and clear. I only have a few comments:

LINE 22: briefly explain what DDE is.

LINE 191: "Among the reference treatments tested, only the HFD-fed larvae and ethanol- 191 exposed larvae showed a significant increase in the steatosis score calculated by our test 192 (Figure 2)." In Figure 2 only the HFD is shown as significative.

LINE 183: please include the following (in bold): "After image acquisition of stained zebrafish larvae with confocal 183 fluorescence microscopy, two types of images were obtained: a first one with a green signal - 184 characteristic of neutral lipid fluorescence – and a second one with a blue signal for normalization 185 – insensitive to neutral lipids" (image A and B, respectively)

A and B should also be indicated in Figure 1b.

LINE 300: "To get further insight into the pathways involved in the development OF steatosis 300 induced by DDE" (include "of") 

If I understand correctly alterations in gene expression were assayed either using the large scale Fluidigm-Biomark technology or by conventional qRT-PCR. The authors should mention the discrepancies found all of the genes analyzed by both assays (e.g. srepf1, pparaa=ppar-alfa?) etc. and potential explanations of this variation.

What was the rational for choosing a 5uM concentration of the SCD1 inhibitor in the experiments described in figure 9?

It would be useful to include the primer conditions for the genes presented in supp. table s4.

.

Author Response

(The authors gave the same response as above.)

Round 2

Reviewer 1 Report

In reply to major commentary #2:

Although the authors now provide justification for the scoring system used in the assay in the methods and with the inclusion of Supplemental Fig.S7, the data encompassing the individual components of the overall StAZ score should nonetheless be included as supplementary data, as presented in the rebuttal Figure A2. Indeed, in the authors’ own reasoned words of the rebuttal letter: “Concerning the integration of these 3 parameters to generate the steatosis score, there is also a rational [sic]. Even if the density of the droplets and the area of droplets could seem more robust than the fluorescence ratio of the Nile red signal to the normalization signal; depending on the nature of steatosis, each one could be important.” (emphasis added). It is justified to the let the readership have access to these components (as supplement) so that each can deem whether the otherwise obfuscated scoring calculation was weighted appropriately.In answer to the major commentary #5:

Indeed not all steatogenic compounds are expected to work via alterations to membrane fluidity, but in the cases where this is observed it would be valuable to know whether the effect is specific to hepatocytes, or whether cell types throughout the animal are similarly affected. This could be addressed by addition of a sentence to the discussion near lines 575-7.

Author Response

We are very grateful to the reviewers for their commentaries regarding first round of revision of our article and their suggestions for improving it. As previously, we have followed their recommendations and have made changes that are described  in point by point answer in attached file.
